# Spatiotemporal Joint Filter Decomposition in 3D Convolutional Neural Networks

**Zichen Miao**[1], **Ze Wang**[1], **Xiuyuan Cheng**[2], and **Qiang Qiu**[1]
Purdue University[1]    Duke University[2]
{miaoz,zewang,qqiu}@purdue.edu    xiuyuan.cheng@duke.edu

## Abstract

In this paper, we introduce spatiotemporal joint filter decomposition to decouple spatial and temporal learning, while preserving spatiotemporal dependency in a video. A 3D convolutional filter is now jointly decomposed over a set of spatial and temporal filter atoms respectively. In this way, a 3D convolutional layer becomes three: a temporal atom layer, a spatial atom layer, and a joint coefficient layer, all three remaining convolutional. One obvious arithmetic manipulation allowed in our joint decomposition is to swap spatial or temporal atoms with a set of atoms that have the same number but different sizes, while keeping the remaining unchanged. For example, as shown later, we can now achieve tempo-invariance by simply dilating temporal atoms only. To illustrate this useful atom-swapping property, we further demonstrate how such a decomposition permits the direct learning of 3D CNNs with full-size videos through iterations of two consecutive sub-stages of learning: In the temporal stage, full-temporal downsampled-spatial data are used to learn temporal atoms and joint coefficients while fixing spatial atoms. In the spatial stage, full-spatial downsampled-temporal data are used for spatial atoms and joint coefficients while fixing temporal atoms. We show empirically on multiple action recognition datasets that, the decoupled spatiotemporal learning significantly reduces the model memory footprints, and allows deep 3D CNNs to model high-spatial long-temporal dependency with limited computational resources while delivering comparable performance.

## 1   Introduction

Convolutional Neural Networks (CNNs) have been used intensively in the field of video understanding. Particularly, networks with 3D convolutional layers capture spatiotemporal correlation and achieve great success in applications like action recognition [1, 30]. However, joint spatiotemporal modeling of videos usually requires significant training time, computation, and memory, which becomes one of the main obstacles in video understanding. In practice, we almost always need to first downsample video data spatially or temporally or both to meet real-world constraints of computational resources and training time. For instance, a 64-frame video with the spatial resolution $224 \times 224$ is downsampled 2 times temporally for training Non-Local Networks [30], and 8 times temporally for the R(2+1)D models [26]. As in most cases, for spatiotemporal modeling of high-resolution videos, 3D CNNs cannot be fit into most modern GPUs due to the huge model memory footprints. However, it has been observed that the full spatial and temporal resolution are essential to achieve superior performance in many video understanding tasks [24, 27, 30].

In this paper, we propose spatiotemporal joint filter decomposition to decouple spatial and temporal learning while preserving spatiotemporal dependency in a video. As shown in Figure 1, a 3D convolutional filter is jointly decomposed over a group of spatial atoms and temporal atoms. The two groups of atoms can reconstruct the 3D filter together with the joint coefficients through tensor multiplication.

35th Conference on Neural Information Processing Systems (NeurIPS 2021).

A single 3D convolutional layer is now divided into three convolutional layers: a temporal-atom layer to focus on time, a spatial-atom layer for space, and a joint coefficient layer to model the spatiotemporal dependency. Thus, spatial and temporal atoms can now be optimized separately. Different from methods that decorrelate the spatial and temporal modeling, the proposed decomposition can still capture spatiotemporal correlations in the joint coefficients. Moreover, our approach can also significantly reduce model parameters and computation complexity by limiting the number of atoms.

One obvious arithmetic property of our joint decomposition is to allow spatial or temporal atoms to be swapped with a set of atoms with the same number but different sizes while keeping

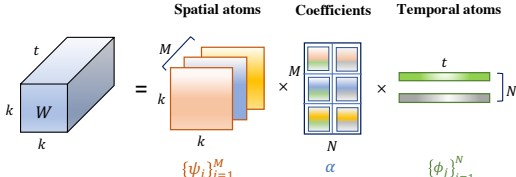

Figure 1: Spatiotemporal jointly decomposed convolutional filters (STDCF) over spatial atoms $\{\psi_i\}_{i=1}^M$, temporal atoms $\{\phi_j\}_{j=1}^N$, and joint coefficients $\boldsymbol{\alpha}$, with $M = 3, N = 2$. It divides one spatiotemporal convolutional layer into three convolutional layers: a spatial-atom layer, a temporal-atom layer, and a joint coefficient layer which mixes information from spatial-atom and temporal-atom convolutional to capture joint dependency.

the remaining unchanged. For example, we can now achieve tempo-invariance by simply dilating temporal atoms only. To exploit this useful atom-swapping property, we further show how the proposed spatiotemporal jointly decomposed convolutional filter (STDCF) permits the direct learning of 3D CNNs with full-size videos.

We start with a pedagogical learning decoupling to train a 3D CNN iteratively with full-temporal downsampled-spatial data first, and then with full-spatial downsampled-temporal data. Note that, such simple-minded decoupled learning is less capable of modeling spatiotemporal dependency. A regular 3D CNN model trained from the two-stage strategy above may capture rich temporal features in the first stage, however, such features will be severely degraded by the downsampled-temporal data in the second stage.

Exploiting the atom-swapping property of the proposed spatiotemporal joint decomposition, we can instead decouple spatial and temporal learning into iterations of two consecutive sub-stages of learning: temporal-focus stage (*stage-t*) and spatial-focus stage (*stage-s*). In *stage-t*, temporal atoms and joint coefficients are learned from full-temporal downsampled-spatial videos by keeping spatial atoms fixed, while in *stage-s*, spatial atoms and coefficients are updated from full-spatial downsampled-temporal data by keeping temporal atoms fixed. Note that, in *stage-s*, spatial atoms are learned from full spatial-resolution, while temporal atoms are frozen to prevent degradation from temporal down-sampling, similar for atoms update in *stage-t*. The proposed two-stage 3D CNN training strategy can model a full temporal and spatial resolution video only from downsampled data while preserving spatiotemporal dependency. Empirically, we show on multiple action recognition datasets, e.g., KTH [19], Kinetics-400 [1] and Something-Somethingv1 [10], that, the STDCF model trained with the proposed strategy significantly reduces memory usage, while producing comparable results with the state-of-the-art models trained with full-size videos.

## 2 Method

In this section, we start by introducing the proposed spatiotemporal jointly decomposed convolutional filter, STDCF, along with its two properties of capturing spatiotemporal dependency and enabling atom swapping. Then, we decouple spatiotemporal learning into iterative two sub-stage learning. Next, we introduce a simple but effective design, enabled by the atom swapping property, to allow the model to learn from videos with different spatial and temporal resolutions in different sub-stages.

### 2.1 Spatiotemporal Joint Filter Decomposition

A regular spatiotemporal convolutional filter in deep video models consists of a group of 3D tensors with shape $t \times k \times k$ for capturing local joint dependency on space and time, where $t$ and $k$ are temporal and spatial filter sizes. The 3D structure not only leads to a dramatic increase in parameters compared to 2D CNNs, but also couples the spatial and temporal learning as each tensor attends to both dimensions simultaneously. To decouple spatial and temporal learning, as well as to reduce model parameters, we propose to jointly decompose a 3D filter over spatial and temporal atoms, as shown in Figure 1. Our idea originates from spatial filter decomposition as proposed in [17], and we extend it to 3D filters by taking the time dimension into consideration.

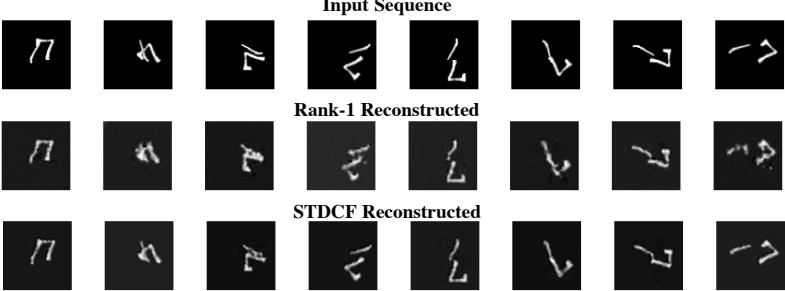

Figure 2: Translate-Rotate-MNIST reconstruction results.

As proposed in [17], a 2D spatial filter $W \in \mathbb{R}^{C_{out} \times C_{in} \times k \times k}$ ($C_{out}$, $C_{in}$ are input, output channels, and $k$ is spatial kernel size), can be decomposed as a linear combination of $M$ spatial atoms $\{\psi_i\}_{i=1}^M, \psi_i \in \mathbb{R}^{k \times k}$, with coefficients $\boldsymbol{\alpha} \in \mathbb{R}^{M \times C_{out} \times C_{in}}$. The spatial convolution then becomes two: A spatial-atom convolution with $\psi_i$, a $1 \times 1$ convolution with joint coefficients $\boldsymbol{\alpha}$. This decomposition not only reduces the parameters and computation cost but also imposes low-rank filter structures.

Then, for a 3D spatiotemporal filter $W \in \mathbb{R}^{C_{out} \times C_{in} \times t \times k \times k}$ ($t$ is the temporal kernel size), as shown in Figure 1, we decompose it over spatial atoms $\Psi = \{\psi_i\}_{i=1}^M$ and temporal atoms $\boldsymbol{\Phi} = \{\phi_j\}_{j=1}^N$, $\phi_j \in \mathbb{R}^t$, with joint coefficients $\boldsymbol{\alpha} \in \mathbb{R}^{C_{out} \times C_{in} \times M \times N}$ as $W = \sum_{i=1}^M \sum_{j=1}^N \boldsymbol{\alpha}^{i,j} \psi_i \phi_j$.

In this way, the 3D convolution becomes three,

$$
\begin{aligned}
\textit{Spatial-atom convolution:} \quad & Z_i(\lambda', t) = \sum_{u'} I(\lambda', t, u + u') \psi_i(u'), \\
\textit{Temporal-atom convolution:} \quad & Y_{ij}(\lambda') = \sum_{t'} \phi_j(t') Z_i(\lambda', t + t'), \\
1 \times 1 \, \textit{joint coefficient convolution:} \quad & J(\lambda, t, u) = \sum_{\lambda'} \sum_{i,j} \alpha^{i,j}_{(\lambda', \lambda)} Y_{ij}(\lambda'),
\end{aligned} \tag{1}
$$

where $I, J$ denotes the input and output, $u \in \mathbb{R}^2, t \in \mathbb{R}, \lambda, \lambda'$ are spatial and temporal indices, $\lambda, \lambda'$ are output, input channel indices, and $Z_i, Y_{ij}$ are intermediate outputs of *spatial-atom layer* and *temporal-atom layer*.

### 2.1.1 Spatiotemporal Dependency

Note that a popular alternative way of 3D filter decomposition is to apply rank-1 decomposition to obtain a 2D spatial filter and a 1D temporal filter, as proposed in [18, 26, 37]. Specifically, a 3D filter $W \in \mathbb{R}^{C_{in} \times C_{out} \times t \times k \times k}$ is decomposed into $W_s \in \mathbb{R}^{C_{in} \times C_{out} \times 1 \times k \times k}$ and $W_t \in \mathbb{R}^{C_{out} \times C_{out} \times t \times 1 \times 1}$ as $W = W_s \times W_k$. In this way, 3D convolution becomes two: spatial sub-convolution with $W_s$, and temporal sub-convolution with $W_t$.

However, such a rank-1 decomposition neglects joint spatiotemporal dependency. The spatial sub-convolution only focuses on spatial modeling, while the temporal sub-convolution merely attends to temporal modeling. The spatial and temporal features are extracted independently by omitting spatiotemporal correlation. Instead, in our STDCF, joint coefficients $\boldsymbol{\alpha}$ encode spatiotemporal dependency while keeping learning in both dimensions independent. We derive below that STDCF has more capacity than rank-1 decomposition. Recall our joint decomposition of 3D filter $W$,

$$
W_{(\lambda, \lambda')} = \sum_{i,j} \boldsymbol{\alpha}^{i,j}_{(\lambda, \lambda')} \psi_i(u) \phi_j(t), \quad \boldsymbol{\alpha}^{i,j}_{(\lambda, \lambda')} \in \mathbb{R}, \tag{2}
$$

$\lambda \in [C], \lambda' \in [C'], i \in [M], j \in [N]$, and $\boldsymbol{\alpha}^{i,j}_{(\lambda, \lambda')}$ are freely trainable. For each fixed pair of $(\lambda, \lambda')$, $\{\boldsymbol{\alpha}^{i,j}_{(\lambda, \lambda')}\}_{i,j}$ is a $M$-by-$N$ matrix, and generally is full rank.

In the rank-1 decomposition, spatial filters are $W_s = W^{(\lambda'', \lambda')}(u)$, temporal filters are $W_t = W^{(\lambda, \lambda'')}(t), \lambda'' \in [C'']$. We can write $W_s^{(\lambda'', \lambda')}$ and $W_t^{(\lambda, \lambda'')}$ as combination of atoms $\psi_i$ and $\phi_j$

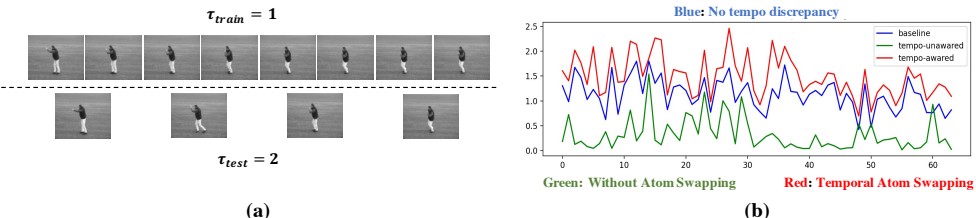

Figure 3: Tempo discrepancy experiments on KTH.

respectively,

$$W_s^{(\lambda'',\lambda')} = \sum_i \mu_{(\lambda'',\lambda')}^i \psi_i, \quad W_t^{(\lambda,\lambda'')} = \sum_j \nu_{(\lambda,\lambda'')}^j \phi_j.$$

Tracking the degree of freedom reveals that rank-1 decomposition is more restrictive than our STDCF. Let $C = C' = C''$, $\boldsymbol{\alpha}$ in STDCF has $MNC^2$ many variables if all free. In comparison, $\boldsymbol{\mu}$ has $MC^2$ many variables, and $\boldsymbol{\nu}$ has $NC^2$ many, thus the rank-1 decomposed convolution formulation has only $(M + N)C^2$ many free variables in total. This quantifies the loss of expressiveness from STDCF to rank-1 decomposition. A more comprehensive analysis is provided in Appendix A.

To empirically illustrate STDCF is more expressive than rank-1 decomposition, we design a toy example on the translate-rotate-MNIST (TR-MNIST) dataset. It consists of short clips of moving digits, which contain complex spatiotemporal correlations. Adopting a 3-layer 3D CNN, we conduct video reconstruction experiments and compare the qualitative and quantitative results between STDCF and rank-1 decomposition. Details of experimental settings are provided in Appendix B. As shown in Figure 2, given a sequence of moving-rotating digits, STDCF reconstructs in better quality than rank-1. As for qualitative results, we measure mean PSNR on the test set. STDCF achieves a mean PSNR of **24.60**, whereas the result of rank-1 decomposition is 21.50, indicating STDCF possesses more capacity to capture spatiotemporal correlations.

### 2.1.2 Atom Swapping Property

The proposed joint decomposition further enables us to manipulate the atoms by changing their spatial or temporal sizes, while keeping their numbers and the rest of components fixed. In this way, we can handle variances in a specific domain by adapting only the corresponding atoms, while keeping the knowledge about spatiotemporal correlation and another domain intact. Rather, we would have to modify the whole 3D filters unwieldy. We call this kind of atom manipulations *Atom Swapping*. We illustrate next its potential usage with a toy tempo-invariance example.

In the temporal domain of videos, a common variance is tempo changes $\tau$, which characterizes the moving speed of objects in a video. We create a toy example based on KTH [19] to illustrate this in Figure 3(a), where the tempo discrepancy happens between training clips ($\tau_{train} = 1$) and test clips ($\tau_{test} = 2$). This leads to test clips proceeding faster than training clips. Without atom swapping, the representation of the model can suffer from severe distortions on the testing set, as shown in Figure 3(b). To tackle this, we can swap temporal atoms with the ones having the corresponding dilation. As shown in Figure 3(b), the model with swapped temporal atoms almost restores the representation compared to the baseline. We provide quantitative results for dilated atoms in Section 3.1.

### 2.2 Decoupled Spatiotemporal Learning

By exploiting the atom swapping property above, we will illustrate next how the proposed STDCF enables direct learning of 3D CNNs with full-size videos. Training deep video models usually utilizes large-scale video datasets with samples in the size of $T \times H \times W$. When $T$ or $H, W$ are large enough, model training will require a significantly large amount of memory. The proposed STDCF jointly decomposes 3D filters as well as 3D convolution, and thus decouples spatial and temporal learning, while preserving dependency across both dimensions. As shown in Figure 4, decoupled learning is performed as iterations of two consecutive sub-stages of learning to model full-size videos but from down-sampled data only, to significantly reduce the memory usage in training.

**Temporal-Focus Learning (*stage-t*).** In this stage, we focus on modeling the temporal dimension using full-temporal downsampled-spatial videos by a factor $\beta$. The original temporal sampling rate is adopted here to update temporal atoms $\{\phi_j\}_{j=1}^N$ to best encode the temporal knowledge. The

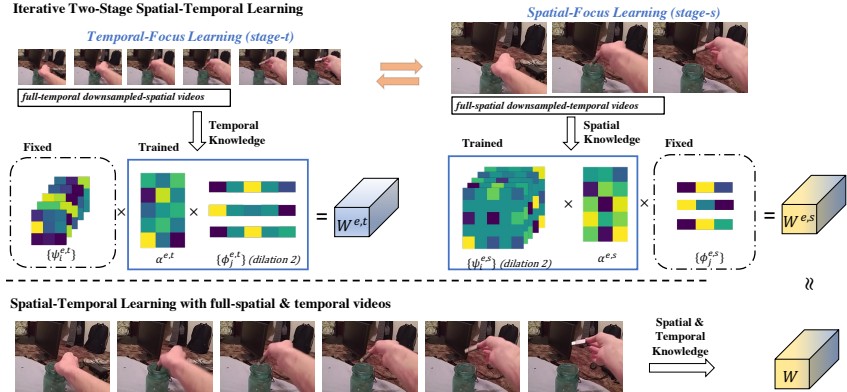

Figure 4: Iterative two-stage spatiotemporal learning. By decomposing 3D filter $W$, we convert spatiotemporal learning from full spatial and temporal resolution videos, which is usually infeasible due to memory issues, to iterations of temporal-focus learning from full-temporal downsampled-spatial videos, and spatial-focus learning from full-spatial downsampled-temporal videos that dramatically reduces the memory footprints. The superscripts of atoms and coefficients denote the current iteration ($\mathbf{e}$) and the sub-stage of learning ($\mathbf{t}$ for *stage-t*, and $\mathbf{s}$ for *stage-s*). Note that our joint decomposition allows us to swap atoms with different dilation to handle resolution changes, e.g. $\{\psi_j^{e,t}\}$ has dilation 2 to deal with full-temporal data, whereas $\{\psi_j^{e,s}\}$ has dilation 1 for 2x downsampled-temporal data.

joint coefficients $\boldsymbol{\alpha}$ are also updated to capture the spatiotemporal dependency. As we downsample training video spatially by a factor $\beta$ to meet the memory limit, we freeze spatial atoms $\{\psi_i\}_{i=1}^M$ to preserve knowledge learned on full spatial-resolution videos in the spatial-focus learning stage next.

**Spatial-Focus Learning (*stage-s*).** Similarly, in this stage, we focus on modeling the spatial dimension using full-spatial downsampled-temporal videos by a factor of $\gamma$. The full spatial resolution is used to update spatial atoms $\{\psi_i\}_{i=1}^M$ to best encode the spatial knowledge, while temporal atoms $\{\phi_j\}_{j=1}^N$ are frozen to preserve knowledge learned on full frame-rate videos in *stage-t* above. The joint coefficient $\boldsymbol{\alpha}$ is updated about the spatiotemporal dependency.

**Iterative Two-Stage Learning.** The above two stages are iterated to model full temporal and spatial resolution videos from down-sampled ones, with far fewer memory footprints, as elaborated in Algorithm 1. Intuitively, more temporal knowledge gained in *stage-t* will further help spatial learning in the subsequent *stage-s* and refine spatiotemporal dependency in joint coefficients. Such iterative two-stage learning (*ITSL*) is empirically validated in Section 3.1.

We provide an interpretation of our iterative two-stage learning from the view of alternating optimization. After being jointly decomposed over spatial and temporal atoms, a 3D filter $W$ becomes $W = \boldsymbol{\Psi}\boldsymbol{\alpha}\boldsymbol{\Phi}^T$, where we use matrix notation here for simplicity, spatial atoms $\Psi \in \mathbb{R}^{k^2 \times M}$, and temporal atoms $\boldsymbol{\Phi} \in \mathbb{R}^{t \times N}$, and the columns of $\Psi$ ($\boldsymbol{\Phi}$) consist of the $M$ ($N$) spatial (temporal) atoms. The atoms $\Psi$ and $\boldsymbol{\Phi}$, and the joint coefficients $\boldsymbol{\alpha}$ are all trainable, and in our proposed algorithm, we train the triplet in an alternative fashion. In the *stage-s*, we fix temporal atoms $\boldsymbol{\Phi}$, and optimize $\Psi$ and $\boldsymbol{\alpha}$ jointly, which is equivalent to training in a spatial CNN; In the *stage-t*, we fix the spatial atoms $\Psi$, and optimize $\boldsymbol{\Phi}$ and $\boldsymbol{\alpha}$ jointly, and this is equivalent to training in a temporal CNN. As a result, the optimization in *stage-s* prepares an improved model for the *stage-t*, and vice versa, and the alternative optimization converges suppose each step can make progress. Combined with our dilated atoms discussed next, in each step we adjust the sampling rate of the data accordingly so as to most efficiently use the memory. The alternative stages store the model refinement in the shared $\boldsymbol{\alpha}$, which is passed to the next stage, and $\boldsymbol{\alpha}$ is independent of geometrical dilation or filter decomposition in space or time.

### 2.3 Dilated Atoms

In ITSL, a severe problem lies in the resolution variation in different sub-stages. Considering the spatial dimension as an example, videos with downsampled spatial resolution, e.g. $\frac{H}{2} \times \frac{W}{2}$ are used in *stage-t*, whereas in *stage-s* and final testing, full spatial size $H \times W$ videos are used; similar for different temporal resolution across sub-stages. We observe this resolution discrepancy in learning and testing will lead to significant performance degradation, which poses a great challenge to our proposed learning strategy. For instance, a model learned on *stage-t* will perform poorly on data in *stage-s* due to the mismatched resolutions, e.g. $T$ v.s. $\frac{T}{2}$.

---

**Algorithm 1** Iterative Two-Stage Learning

---

**Require:**

Full-size video training dataset $D = \{(\boldsymbol{x_i}, y_i)\}_{i=1}^N$, $\boldsymbol{x_i}$ of size $T \times H \times W$

Initialize the STDCF model $\mathcal{M}$ with spatial atoms $\{\psi_i\}_{i=1}^M$, temporal atoms $\{\phi_j\}_{j=1}^N$, joint coefficient $\boldsymbol{\alpha}$

Classification loss function $L$. Iterations $E$.

$\psi_i^{0,s}, \phi_j^{0,s}, \boldsymbol{\alpha}^{0,s} \leftarrow \psi_i, \phi_j, \boldsymbol{\alpha}$

**for** $e = 1, 2, ..., E$ **do**

    *stage-t* **Training**:

    Acquire full-temporal downsampled-spatial data: $D_s = \{(\boldsymbol{x_i'}, y_i)\}_{i=1}^N$, $\boldsymbol{x_i'}$ of size $T \times \frac{H}{\beta} \times \frac{W}{\beta}$

    $\phi_j^{e,t}, \boldsymbol{\alpha}^{e,t} \leftarrow \nabla_{\mathcal{M}} L(\mathcal{M}(D_s | \psi_i^{e-1,s}, \phi_j^{e-1,s}, \boldsymbol{\alpha}^{e-1,s}), \psi_i^{e,t} \leftarrow \psi_i^{e-1,s}$

    *stage-s* **Training**:

    Acquire full-spatial downsampled-temporal data: $D_t = \{(\boldsymbol{x_i''}, y_i)\}_{i=1}^N$, $\boldsymbol{x_i''}$ of size $\frac{T}{\gamma} \times H \times W$

    $\psi_i^{e,s}, \boldsymbol{\alpha}^{e,s} \leftarrow \nabla_{\mathcal{M}} L(\mathcal{M}(D_t | \psi_i^{e,t}, \phi_j^{e,t}, \boldsymbol{\alpha}^{e,t}), \phi_j^{e,s} \leftarrow \phi_j^{e,t}$

**end for**

**Return** $\mathcal{M}(\psi_i^{E,s}, \phi_j^{E,s}, \boldsymbol{\alpha}^{E,s})$

---

As discussed in Section 2.1.2, we can handle this problem using dilated atoms. To be specific, model learned in *stage-t* from videos of size $T \times \frac{H}{\beta} \times \frac{W}{\beta}$, has its spatial atoms $\{\psi_i\}_{i=1}^M$ dilated by factor $\beta$ when learned in *stage-s* and tested on full spatial-resolution videos. Similarly, temporal atoms $\{\phi_j\}_{j=1}^N$ will be dilated by $\gamma$ when the model is trained in *stage-t* and tested, and dilated by 1 in *stage-s*.

## 2.4 Parameter and Computation Reduction

Regular 3D filters have a significant increase in parameters and computation flops compared to 2D filters, which contributes to the huge demand on memory and computation power. The proposed STDCF reduces the parameters and computation complexity of 3D filters significantly.

To be specific, a regular 2D filter is of size $C_{in} \times C_{out} \times k \times k$ has $C_{in}C_{out}k^2$ parameters, while regular 3D filter has $C_{in}C_{out}tk^2$ parameters with the additional time dimension. In STDCF, the joint coefficient $\boldsymbol{\alpha}$ has $C_{in}C_{out}MN$ parameters, and spatial and temporal atoms $\{\psi_i\}_{i=1}^M, \{\phi_j\}_{j=1}^N$ has $Mk^2, Nt$ parameters, respectively. Thus, the reduction rate in number of parameters compared to the original 3D filter is $\frac{Nt+Mk^2+C_{in}C_{out}MN}{C_{in}C_{out}tk^2}$. In practice, $C_{in}C_{out}MN$ is much larger than $Nt + Mk^2$, which gives the approximate reduction of parameters rate of $\frac{N}{t} \cdot \frac{M}{k^2}$ than regular 3D filters, and $\frac{MN}{k^2}$ compared to regular 2D filters. If $t = k = 3$, and we can choose $M = 3, N = 2$, giving a reduction rate of $\frac{2}{9}$ compared to non-decomposed 3D filters, and $\frac{2}{3}$ compared to non-decomposed 2D filters.

As for the computation cost, regular 2D convolution with input with shape of $C_{in} \times W \times W$ needs $C_{out}C_{in}W^2(1 + 2k^2) \approx 2C_{out}C_{in}W^2k^2$ FLOPs, and regular spatiotemporal convolution with inputs in size of $C_{in} \times T \times W \times W$ costs $C_{out}C_{in}W^2T(1 + 2k^2t) \approx 2C_{out}C_{in}W^2Tk^2t$ FLOPs. In STDCF, the computation is split into three sub-layers: (1) spatial-atom layer with $\psi_i$ costs $2Mk^2C_{in}TW^2$ FLOPs, (2) temporalatom layer with $\phi_j$ requires $2NtMC_{in}TW^2$ FLOPs, (3) coefficient layer with $\boldsymbol{\alpha}$ on every spatiotemporal location costs $2C_{out}C_{in}MNW^2T$ FLOPs totally. Together, the computation cost is $2C_{in}MTW^2(k^2 + Nt + NC_{out})$ FLOPs. Since $NC_{out}$ is usually

Table 1: Dilated atoms experiment. $D_s^{te}, D_t^{tr}$ indicate the dilation of spatial/temporal atoms in the testing/learning phase.

| Train size | 16x30x40 | | 16x60x80 | |
|---|---|---|---|---|
| Test size | 16**x60x80** | | **8**x60x80 | |
| Dilation | $D_s^{te} = 1$ | $D_s^{te} = 2$ | $D_t^{tr} = 1$ | $D_t^{tr} = 2$ |
| Acc. | 53.08 | **75.34** | 66.49 | **77.11** |

Table 2: Accuracies and GPU memory footprints of different 3D layers on KTH dataset under regular full-size videos learning and the proposed iterative two-stage learning.

| Layer type | full-size learning | | ITSL ($E = 3$) | |
|---|---|---|---|---|
| | Memory | Acc. | Memory | Acc. |
| Reg. 3D | 459.7M | 81.26 | 229.5M (50.07% ↓) | 76.94 (4.32 ↓) |
| Rank-1 3D | 582.5M | 82.55 | 290.5M (50.13% ↓) | 79.10 (3.45 ↓) |
| STDCF | 459.4M | **85.61** | 228.9M (50.17% ↓) | **84.93 (0.68 ↓)** |

much larger than $k^2 + Nt$, the cost is around $2C_{in}C_{out}TW^2MN$ FLOPs, showing that the reduction rate in computation FLOPs is again $\frac{N}{t} \cdot \frac{M}{k^2}$ compared to regular 3D convolution.

## 3  Experiments

In this section, we validate the proposed approach on multiple action recognition datasets, KTH [19], Kinetics-400 [1], and Something-Somethingv1 [10].

### 3.1  Illustrative Experiments

We illustrate the basic idea of the proposed decomposed learning process on the KTH action recognition dataset [19]. The KTH dataset consists of 600 videos at 25 fps of 6 action classes from 25 people. The videos are further sampled as short clips of size $16 \times 80 \times 60$, which are used in the full-size learning setting. The training set contains 4667 clips, and the validation set has 4551 clips. Models are learned on the training set and report accuracy on the validation set.

We adopt a simple three-layer 3D CNN model with regular 3D convolution (Reg. 3D) as the baseline, and further replace each convolutional layer with STDCF and Rank-1 3D decomposed convolutional layer (Rank-1 3D) in Section 2.1.1, as shown in Table B. For training models with full-size data, we set batch-size as 16, learning rate as 0.001, and train all models for 40 epochs, with learning rate reduced to 0.0001 at the 20th epoch. For ITSL, we set total iterations $E = 3$, $\beta = \gamma = 2$, i.e. we downsample clips spatially to the size of $16 \times 30 \times 40$ in *stage-t*, and temporally to $8 \times 60 \times 80$ in *stage-s*. To counter the discrepancy in resolution, we dilate temporal atoms by 2 in *stage-t*, spatial atoms by 2 in *stage-s*, and both atoms by 2 for testing on full-size clips. We also evaluate the Reg. 3D and Rank-1 3D model under ITSL data setting. For Reg. 3D, we train its 3D filters in both *stage-t* and *stage-s* with downsampled data, and for Rank-1 3D, we train its $W_t$ with full-temporal downsampled-spatial videos, $W_s$ with full-spatial downsampled-temporal videos. Dilated convolutions are applied to these two models with the same dilation setting as STDCF. The training setting aforementioned is adopted to each *stage-t* and *stage-s*, and all models trained with ITSL are tested with full-size clips.

We illustrate the effectiveness of the proposed dilated atoms in Table 1. A STDCF model is learned and tested on data with resolution discrepancy. The poor performance caused by the spatial discrepancy can be improved by dilating spatial atoms during testing. Same for the temporal dimension, dilating temporal atoms can counter the degradation caused by variation in temporal resolution.

Then, as shown in Table 2, we report accuracies on validation clips and the maximum training GPU memory usage with the above settings. Specifically, in ITSL experiments, we report the max memory usage in one iteration, which occurs in *stage-s*. Under the full-size learning setting, STDCF gives better performance over the Reg. 3D and the Rank-1 3D, showing its effectiveness in capturing spatiotemporal dependency. In the ITSL setting, the GPU memory used is significantly reduced for all models (around 50%). However, the performance of Reg. 3D model drops by a large margin because of the downsampled training clips as well as the coupled spatial and temporal learning. The Rank-1 3D model decouples spatial and temporal learning but still shows a large accuracy drop as it can not fully leverage the iterative learning. On the

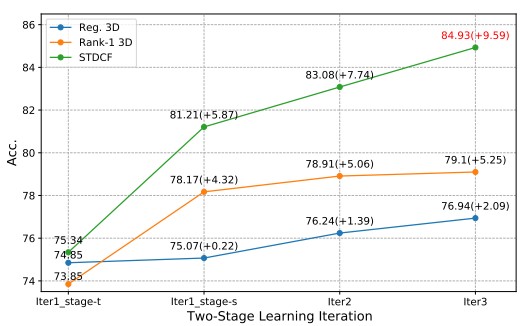

Figure 5: Accuracies of different models after every iteration of two-stage spatiotemporal learning. The number in the parenthesis is the relative performance gain w.r.t. the model trained after *stage-t* in the first iteration.

other hand, STDCF with ITSL gives the closest accuracy to the one obtained in full-size learning by a small difference of $0.68\%$, validating that it not only decouples spatial and temporal learning but also refines spatiotemporal modeling iteratively.

We further plot accuracies of different models at each learning iteration, as shown in Figure 5. Similarly, due to coupled spatial and temporal learning, Reg. 3D model only gains $0.22$ in performance in *stage-s* compared to *stage-t* in the first learning iteration, and its performance does not increase significantly (by $1.17, 0.7$) in the two learning iterations afterward. As for the Rank-1 3D model, though it shows a large increase from Iter_stage-t to Iter_stage-s in Iter1, it does not produce a substantial gain in accuracy in Iter2, Iter3. In comparison, the proposed STDCF encodes spatial and

Table 3: Results on Kinetics-400. We only compare with methods with the same setting as ours.

| Model | #Params | **Memory** | Top-1 | Top-5 | GFLOPs |
|---|---|---|---|---|---|
| R(2+1)D [26] | 63.7M | 5.0G | 73.1 | 89.8 | 53.2 |
| SlowFast [8] | 34.6M | 9.1G | 75.6 | 92.3 | 65.7 |
| TSM [15] | 24.3M | 7.1G | 74.1 | 90.8 | 33.0 |
| TIN [21] | 24.3M | 6.2G | 70.9 | 89.8 | 34.0 |
| STDCF-R50-t-3 | | **1.9G** | 73.6 | 90.3 | |
| STDCF-R50-s-3 | 18.5M | **3.8G** | 74.0 | 90.6 | 52.2 |
| STDCF-R50 | | 7.9G | 74.5 | 91.2 | |

temporal knowledge to the corresponding atoms in *stage-t* and *stage-s* separately, and thus shows a significant leap in accuracy from Iter_stage-t to Iter_stage-s. Besides, as the joint coefficient $\alpha$ captures richer spatiotemporal dependency with the model trained iteratively, the STDCF model shows a consistent and substantial performance gain in the next two learning iterations (by $1.87, 1.85$), and an accuracy comparable to full-size learning in the end.

## 3.2 Experiments on Kinetics-400

### 3.2.1 Experimental Setup

Kinetics-400 [1] consists of around 240k training videos, 19k validation videos for total 400 action classes. All experiments are conducted with RGB modality and evaluated on validation sets. Following the settings in [8], input frames are sampled temporally from 64 consecutive frames at an interval of 8, which is the full-temporal size. Spatially, each frame is randomly rescaled so that its short side is in [256, 320], and then randomly flipped, cropped into $224 \times 224$, which is the full-spatial size in our ITSL setting. As for ITSL setting, we set $E = 3$, $\beta = \gamma = 2$, so STDCF-R50 is learned with videos of the size $8 \times 112 \times 112$ in *stage-t*, and with videos of the size $4 \times 224 \times 224$ in *stage-s*.

We adopt 3D ResNet-50 (3D-R50) as the backbone, which inflates $3 \times 3$ kernels in 2D ResNet-50 [11] to $3 \times 3 \times 3$. We further substitutes all those spatiotemporal filters with our STDCF, where we set $M = 6, N = 2$, and obtain STDCF-R50. The architecture of STDCF-R50 is shown in Table C. STDCF-R50 is learned from scratch for 256 epochs in *stage-t* of the first iteration ($e = 1$) with the SGD optimizer (momentum of 0.9) of learning rate of 0.03, and weight decay of 0.0001. In *stage-s* and *stage-t* of $e = 2, 3$, STDCF-R50 is learned with the same hyper-parameters as in *stage-t* of the first iteration except for learning rate reduced to 0.0005, and updated for 50 epochs. For inference, we adopt the *30-crop* setting as in [8] to conduct the inference with full-size videos. Specifically, we uniformly sample 10 clips from a video along its temporal dimension. Then, we downsample each frame spatially so that its short side size is 256, and take three crops of $256 \times 256$ covering all spatial information. We dilate both temporal atoms and spatial atoms by 2 in inference for STDCF-R50-t and STDCF-R50-s.

### 3.2.2 Results

We firstly compare STDCF-R50 learned in *stage-t*, STDCF-R50-t-3, with the one in *stage-s* that has updated spatial atoms, STDCF-R50-s-3 of the final iteration ($E = 3$). As shown in Table 3, the performance gain in STDCF-R50-s-3 w.r.t. STDCF-R50-t-3 indicates the temporal knowledge and spatial knowledge are learned separately without mutual interference, validating the effectiveness of the proposed ITSL. We further compare STDCF-R50-s-3 with the model learned from full-size data, STDCF-R50. The only 0.5% performance gap shows the proposed model learned with iterative learning strategy can achieve nearly the same performance of the full-size learned model with a half GPU memory usage reduction. The accuracies of models in all iterations are provided in Table D. Then, we compare our models with other state-of-the-art models. Note that for a fair comparison, we only compare with methods using networks of the same depth, and the same data setting as ours. STDCF-R50 learned with ITSL shows comparable results with other methods, with significantly fewer GPU memory footprints in the learning phase. Moreover, we evaluate the number of parameters and computation FLOPs per clip in the inference phase. Compared to other methods, STDCF models show a significant reduction in the number of parameters and comparable FLOPs.

Table 4: Results on Something-Somethingv1

| Method | Memory | Pretrain | # frames | Val Top-1 | Val Top-5 |
|--------|--------|----------|----------|-----------|-----------|
| 2D-R50 | 5.9G | ImageNet | 8 | 22.3 | 48.2 |
| TSN [29] | - | ImageNet | 16 | 19.7 | 46.6 |
| TRN [40] | - | ImageNet | 8 | 34.4 | - |
| I3D [1] | - | Kinetics | 32 | 41.6 | 72.2 |
| I3D+GCN [31] | - | Kinetics | 32 | 43.3 | 75.1 |
| TIN [21] | 7.5G | Kinetics | 8 | 45.8 | 75.1 |
| TSM [15] | 7.1G | ImageNet | 8 | 45.6 | 74.2 |
| STDCF-R50-t-2 | **1.9G** | Kinetics | 8 | 44.8 | 74.3 |
| STDCF-R50-s-2 | **3.8G** | - | 4 | 45.1 | 74.7 |
| STDCF-R50 | 7.8G | Kinetics | 8 | **45.9** | **75.2** |

## 3.3 Experiments on Something-Somethingv1

Something-Somethingv1 [10] is a more challenging dataset, as the activity can only be inferred by modeling spatiotemporal joint dependency. It contains 86k training videos and 12k validation videos for 174 classes. We apply the same model STDCF-R50 with pretrained on Kinetics-400. The model learned directly from full-size videos is finetuned for 50 epochs with batchsize of 64, learning rate of 0.001 which drops by 0.1 at the 30th epoch. For the ITSL setting, we set $E = 2$, $\beta = \gamma = 2$. Models learned with ITSL are all finetuned for 20 epochs with learning rate of 0.0002 which drops by 0.1 at the 15th epoch.

As shown in Table 4, our full-size learned model, STDCF-R50, achieves competitive results with other methods. Moreover, our model learned with ITSL can nearly recover the full model's performance, while reducing the GPU memory usage by half. The results of two iterations of ITSL are illustrated in Table E.

## 3.4 Visualization

**KTH visualization.** To better understand the proposed decoupled spatiotemporal learning, we visualize the feature maps of the last convolutional layer in KTH experiment, as shown in Figure 6. Note that we sum all feature maps along the channel index, and adds it back to original frames. We compare the model from the first iteration *stage-t* (*Iter-1 stage-t*) and the one from the second iteration *stage-s* (*Iter-2 stage-s*). In both Figure 6a and Figure 6b, the *Iter-1 stage-t* model fail to

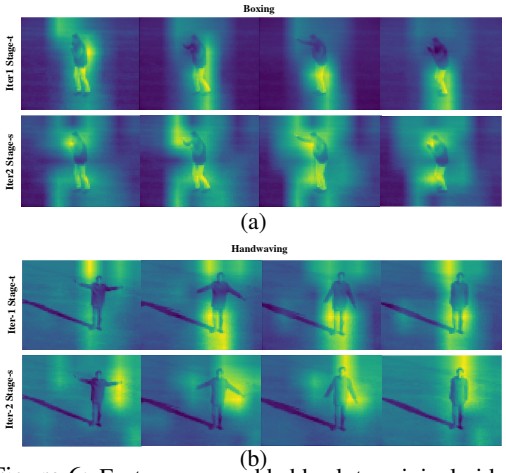

Figure 6: Feature maps added back to original video frames to show where the network attends to. In (a), *Iter-1 stage-t* model does not focus on the **right arm** region while *Iter-2 stage-s* model does. In (b), *Iter-1 stage-t* model does not focus on **right and left arms** regions while *Iter-2 stage-s* model does.

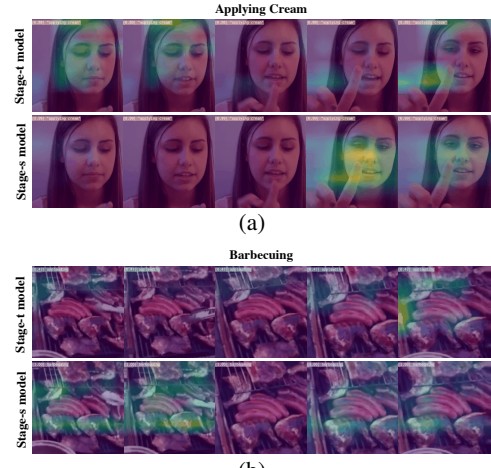

Figure 7: Activation maps of *stage-t* and *stage-s* models that highlights region contributing to the prediction. In (a), both *stage-t* and *stage-s* predict right, but the model fine-tuned in *stage-s* focuses more on the **finger with cream**. In (b), *stage-s* model corrects the wrong prediction in *stage-t* from *applauding* to *barbecuing* by attending to **meats and sausages.**

attend to the discriminative part of frames, i.e., right arm in Figure 6a, two arms in Figure 6b, whereas the *Iter-2 stage-s* does by learning finer spatial knowledge.

**Kinetics visualization.** We further use Grad-CAM [20] to visualize the activation maps of STDCF-R50-t and STDCF-R50-s on Kinetics that highlight region contribution to the prediction, as shown in Figure 7. The top row of frames in Figure 7a illustrates the activation maps of STDCF-R50-t, and the bottom row of frames are for STDCF-R50-s. The frames belong to class 'applying the cream'. It shows that while the *stage-t* model captures moving regions and gets the correct prediction, the learning in *stage-s* refines the activations on space to the most discriminative areas. In Figure 7b, activations for *stage-t* model on the top row attends to wrong regions and makes the wrong prediction. By fine-tuning in *stage-s*, the model focuses on the correct area and corrects its prediction to 'barbecuing'.

## 4    Related Works

**3D CNN Models.** In [13], 2D CNN models are directly extended to videos by temporal pooling all frames' spatial features extracted by 2D CNN per video. However, this method ignores the temporal relations, so [6, 39] replace the temporal pooling with the LSTM to capture high-level spatiotemporal dependency. To model low-level spatiotemporal dependency, 3D CNNs [1, 12, 23, 25, 28] have been proposed and improved the performance on action recognition significantly by capturing richer spatiotemporal correlation. Nevertheless, 3D CNNs achieve this with a dramatic increase in training time, computation, and memory. [18, 26, 37] propose to decompose the 3D filter into a 2D filter to capture spatial dependency, and a 1D filter for temporal dependency, so as to reduce the cost of regular 3D kernels. This decomposition have been applied to recent methods such as Non-local [30], SlowFast [8], and TPN [38]. However, this decomposition loses the ability to jointly model space and time as regular 3D filters, as it performs spatial and temporal modeling sequentially. STDCF, on the other hand, reduces the parameter number and computation cost, while capturing spatiotemporal dependency.

**Efficient Video Models Learning.** Improvements in the efficiency of deep models learning can originate from refined optimization algorithms [7, 14, 16, 22], pre-training [5], and softwares for acceleration [4]. For deep video model's learning, pre-training schemes [1, 9] have been widely adopted. Recently proposed Multigrid [36] has further reduced the amount of training time of deep video models. The above methods mainly focus on learning efficiency in terms of time. We propose a complementary method that attends to the reduction of memory footprints, by avoiding the usage of full-size samples.

**Decomposed Convolutional Filters.** Decomposed filters is proposed in [17] for parameter and computation efficiency with provably representation stability. It is further extended to domain adaptation [34], generative modeling [32], graph neural network [2], equivariant networks [3, 41], and flexible convolution [33, 35]. In this paper, we extend [17] to 3D filter which is jointly decomposed over spatial and temporal atoms for decoupling spatiotemporal learning.

## 5    Conclusion

In this paper, we proposed to decouple spatiotemporal learning to significantly reduce the model size, computation, and memory usage in modeling videos. We adopted an iterative two-stage learning to model full-resolution videos with only downsampled data to significantly reduce the memory footprints in the learning phase by around 50%, while producing comparable results with full-size data learning. The decoupled learning paves the way for more accessible and scalable video understanding.

## Acknowledgement

The work is supported by the DARPA TAMI program under No. HR00112190038 and NSF (DMS-1820827). XC is partially supported by NIH and the Alfred P. Sloan Foundation.

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
