# Appendix

## A  Capacity Analysis

In below we give a formal analysis of the higher capacity of STDCF than rank-1 decomposition.

For simplicity, we assume $C_{in} = C_{out} = 1$, and only consider the linear convolution operator (omitting the addition of bias and non-linear activation).

We derive the spatiotemporal convolution in continuous integral over space and time. The regular spatiotemporal joint convolution is denoted as $I \circledast W(u, t) := \int \int I(u + u', t + t')W(u', t')du'dt'$, with a filter $W(u, t)$, where $u \in \mathbb{R}^2$, and $t \in \mathbb{R}$. Then, the joint convolution generally can be written as

$$J(\lambda) = \sum_{\lambda'} I(\lambda') \circledast W_{(\lambda, \lambda')}$$

where $W_{(\lambda, \lambda')} : \mathbb{R}^2 \times \mathbb{R} \to \mathbb{R}$ is local both in space and in time. The proposed convolution by atom is equivalently to writing $W_{\lambda, \lambda'}$ as

$$W_{(\lambda, \lambda')} = \sum_{i,j} \alpha_{(\lambda, \lambda')}^{i,j} \psi_i(u)\phi_j(t), \quad \alpha_{(\lambda, \lambda')}^{i,j} \in \mathbb{R}, \tag{1}$$

and $a_{k,l}^{(c,c')}$ are freely trainable, and then

$$J(\lambda) = \sum_{\lambda'} \sum_{i,j} \alpha_{(\lambda, \lambda')}^{i,j} I(\lambda') \circledast (\psi_i \otimes \phi_j), \tag{2}$$

where $\psi \otimes \phi$ denotes the tensor-ed atom, namely $\psi \otimes \phi(u, t) = \psi(u)\phi(t)$. For each fixed pair of $(\lambda, \lambda')$, $\{\alpha_{(\lambda, \lambda')}^{i,j}\}_{i,j}$ is a $M$-by-$N$ matrix, and generally is full rank.

We compare with applying the space convolution and then the temporal convolution sequentially, i.e. the rank-1 3D filter decomposition. In this decomposition, spatial filters are $W_s = W^{(\lambda'', \lambda')}(u)$, temporal filters are $W_t = W^{(\lambda, \lambda'')}(t)$, $\lambda'' \in [C'']$. Then,

$$Z_{\lambda''} = \sum_{\lambda'} X_{\lambda'} *_u W_s^{(\lambda'', \lambda')}, \quad Y_\lambda = \sum_{\lambda''} Z_{\lambda''} *_t W_t^{(\lambda, \lambda'')}, \tag{3}$$

where $*_u, *_t$ denote spatial and temporal convolution. Write $W_s^{(\lambda'', \lambda')}$ and $W_t^{(\lambda, \lambda'')}$ as combination of atoms $\psi_i$ and $\phi_j$ respectively,

$$W_s^{(\lambda'', \lambda')} = \sum_i m_{(\lambda'', \lambda')}^i \psi_i, \quad W_t^{(\lambda, \lambda'')} = \sum_j n_{(\lambda, \lambda'')}^j \phi_j,$$

then $Y_c$ can also be expressed as (2) where

$$\boldsymbol{\alpha}_{(\lambda, \lambda')}^{i,j} = \sum_{\lambda''=1}^{C''} m_{(\lambda'', \lambda')}^i n_{(\lambda, \lambda'')}^j, \quad i \in [K_u], j \in [K_t].$$

Tracking the degree of freedom reveals that $\boldsymbol{\alpha}_{(\lambda, \lambda')}^{i,j}$ in the above form is more restrictive than being free parameters: To simplify notation, let $C = C' = C''$, then $\boldsymbol{\alpha}_{(\lambda, \lambda')}^{i,j}$ has $C^2 K_u K_t$ many variables if all free. In comparison, $m_{(\lambda'', \lambda')}^i$ has $K_u C^2$ many variables, and $n_{(\lambda, \lambda'')}^j$ has $K_t C^2$ many, thus the rank-1 decomposed convolution formulation has only $(K_u + K_t)C^2$ many free variables in total. This quantifies the loss of expressiveness from (2) to (3).

## B  Details about Translate-Rotate MNIST Reconstruction

Here we provide the details of the translate-rotate mnist reconstruction experiments.

| layer | Rank-1 3D | STDCF |
|---|---|---|
| conv1 | $1 \times 3^2, 4, s(1,2,2)$ 
 $3 \times 1^2, 4, s(2,1,1)$ | $3 \times 3^2, 4, s(2,2,2)$ 
 $(M=5, N=3)$ |
| conv2 | $1 \times 3^2, 8, s(1,2,2)$ 
 $3 \times 1^2, 8, s(2,1,1)$ | $3 \times 3^2, 8, s(2,2,2)$ 
 $(M=5, N=3)$ |
| deconv1 | $1 \times 3^2, 4, s(1,2,2)$ 
 $3 \times 1^2, 4, s(2,1,1)$ | $3 \times 3^2, 4, s(2,2,2)$ 
 $(M=5, N=3)$ |
| deconv2 | $1 \times 3^2, 4, s(1,2,2)$ 
 $3 \times 1^2, 4, s(2,1,1)$ | $3 \times 3^2, 4, s(2,2,2)$ 
 $(M=5, N=3)$ |
| conv3 | $1 \times 3^2, 4, s(1,1,1)$ 
 $3 \times 1^2, 1, s(1,1,1)$ | $3 \times 3^2, 1, s(1,1,1)$ 
 $(M=5, N=3)$ |

Table A: Architectures for Translate-Rotate MNIST reconstruction experiments. s(2, 2, 2) indicates the stride for 3D convolution.

**Dataset.** For training set, we randomly select 20,000 digits from original MNIST training set, and create 10,000 8-frame clips with 2 digits in each. For each clip, two digits start translation in random speeds from random positions, where the a digit will bounce backwards when it hit the border of the frame. The frame size is set to be 28, and the digit is formatted as the original MNIST $28 \times 28$ image. While the digit is translating, it is also rotating in a angular speed of 45 degree/frame to form complex spatiotemporal correlations. In additional, two digits can also overlap to make the reconstruction task more difficult. We construct 5,000 8-frame test clips in the same way of building the training set.

**AutoEncoder Architecture and Training Details.** We adopt a 2-layer 3D CNN for the encoder and 3-layer 3D CNN for the decoder. The autoencoder is instantiated by inserting the rank-1 decomposition or STDCF, as shown in Table A. For Training, we adopt the $L2$ loss, and use Adam optimizer with lr $= 1e - 3$, batchsize 64. We train the model for total 50 epochs.

**Additional Qualitative Results.** We provide additional visualization results to show STDCF captures more spatiotemporal correlations than rank-1 decomposition. As shown in Figure A, STDCF consistently outperforms rank-1 decomposed 3D filters in reconstruction qualities.

## C  Details about the KTH experiments

We provide the architecture we used for KTH in both Section 2.1.2 and Section 3.1 in Table B. The 64-dimension representations shown in Figure 3 are obtained after conv3. the baseline method is the representation with $\tau_{test} = \tau_{train} = 1$. the tempo-awared methods is to use dilation=$(2, 1, 1)$ in all three convolutional layers. We provide more representation samples in Figure B.

Table B: Architectures for KTH experiment

| layer | Reg. 3D | Rank-1 3D | STDCF |
|---|---|---|---|
| conv1 | $5 \times 3^2, 16$ | $1 \times 3^2, 16$ 
 $5 \times 1^2, 16$ | $5 \times 3^2, 16$ 
 $(M=5, N=3)$ |
| | | max-pool 1, 2, 2 | |
| conv2 | $5 \times 3^2, 32$ | $1 \times 3^2, 32$ 
 $5 \times 1^2, 32$ | $5 \times 3^2, 32$ 
 $(M=5, N=3)$ |
| | | max-pool 2, 2, 2 | |
| conv3 | $3 \times 3^2, 64$ | $1 \times 3^2, 64$ 
 $3 \times 1^2, 64$ | $3 \times 3^2, 64$ 
 $(M=5, N=2)$ |
| | | max-pool 2, 2, 2 | |
| | | global average pool, fc | |

## D  Details about the Kinetics and Something-Somethingv1 experiments

### D.1  Architecture

We provide the architecture of STDCF-R50 in Table C.

Table C: Architecture of STDCF-R50.

| Stage | Layer | Output Size |
|---|---|---|
| raw | - | $L \times 224 \times 224$ |
| $\text{conv}_1$ | $5 \times 7 \times 7, 64$, stride $1, 2, 2$ | $L \times 112 \times 112$ |
| $\text{pool}_1$ | $1 \times 3 \times 3$, max, stride $1, 2, 2$ | $L \times 56 \times 56$ |
| $\text{res}_2$ | $\begin{bmatrix} 1 \times 1 \times 1, 64 \\ \text{STDCF } 3 \times 3 \times 3, 64 \\ 1 \times 1 \times 1, 256 \end{bmatrix} \times 3$ | $L \times 56 \times 56$ |
| $\text{res}_3$ | $\begin{bmatrix} 1 \times 1 \times 1, 128 \\ \text{STDCF } 3 \times 3 \times 3, 128 \\ 1 \times 1 \times 1, 512 \end{bmatrix} \times 4$ | $L \times 28 \times 28$ |
| $\text{res}_4$ | $\begin{bmatrix} 1 \times 1 \times 1, 256 \\ \text{STDCF } 3 \times 3 \times 3, 256 \\ 1 \times 1 \times 1, 1024 \end{bmatrix} \times 6$ | $L \times 14 \times 14$ |
| $\text{res}_5$ | $\begin{bmatrix} 1 \times 1 \times 1, 512 \\ \text{STDCF } 3 \times 3 \times 3, 512 \\ 1 \times 1 \times 1, 2048 \end{bmatrix} \times 3$ | $L \times 7 \times 7$ |
| | global average pool,fc | $1 \times 1 \times 1$ |

### D.2 Accuracies of all ITSL iterations on Kinetics

We provided accuracies of all models learned in *stage-t* and *stage-s* of all three iterations on Kinetics-400 in Table D.

Table D: Accuracies of *stage-t* and *stage-s* models of all three iterations.

| Method | Top-1 Acc. | Top-5 Acc. |
|---|---|---|
| STDCF-R50-t-1 | 68.2 | 88.4 |
| STDCF-R50-s-1 | 70.8 | 89.1 |
| STDCF-R50-t-2 | 72.0 | 89.7 |
| STDCF-R50-s-2 | 73.1 | 90.2 |
| STDCF-R50-t-3 | 73.6 | 90.3 |
| STDCF-R50-s-3 | 74.0 | 90.6 |
| STDCF-R50 | 74.5 | 91.2 |

### D.3 Accuracies of all ITSL iterations on Something-Somethingv1

We provided accuracies of all models learned in *stage-t* and *stage-s* of all three iterations on Something-Somethingv1 in Table E.

Table E: Accuracies of *stage-t* and *stage-s* models of two iterations.

| Method | Top-1 Acc. | Top-5 Acc. |
|---|---|---|
| STDCF-R50-t-1 | 42.3 | 71.8 |
| STDCF-R50-s-1 | 44.1 | 73.6 |
| STDCF-R50-t-2 | 44.8 | 74.3 |
| STDCF-R50-s-2 | 45.1 | 74.7 |
| STDCF-R50 | 45.9 | 75.2 |

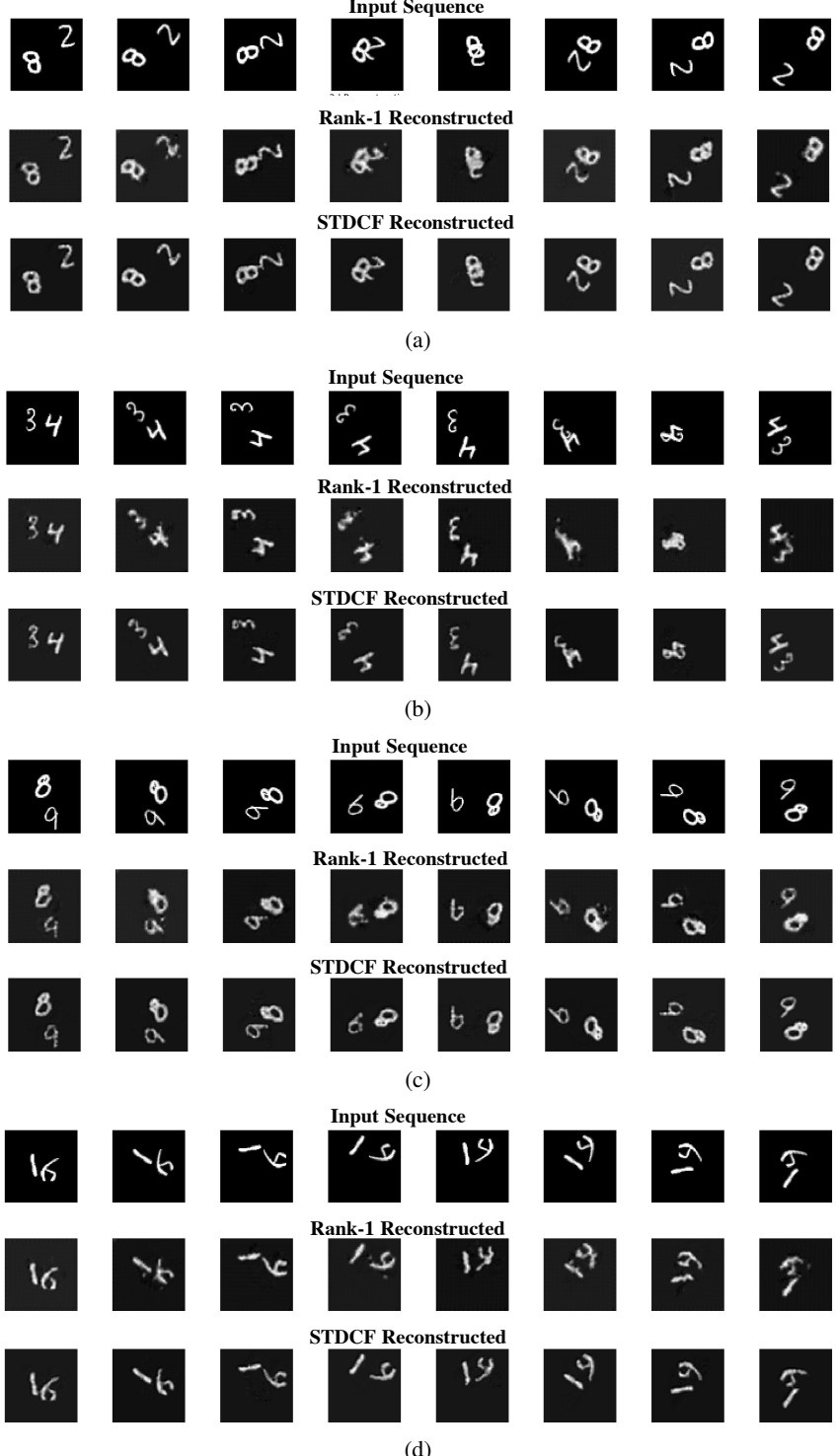

Figure A: More visualizations for TR-MNIST reconstruction.

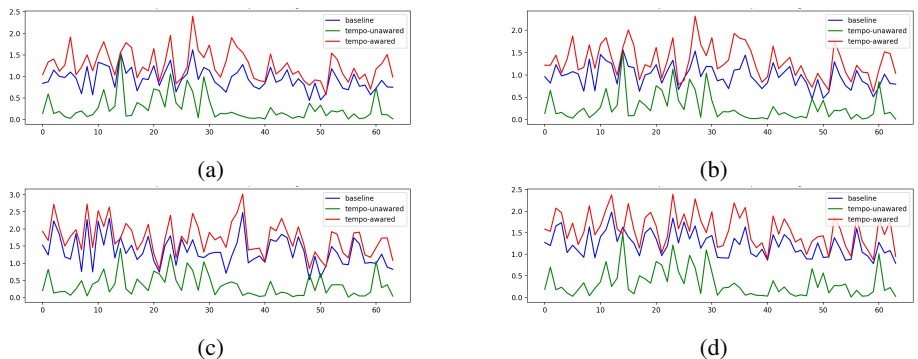

Figure B: More visualizations of representation comparisons.