# OpenReview forum: "Spatiotemporal Joint Filter Decomposition in 3D Convolutional Neural Networks"
_NeurIPS.cc/2021/Conference — NeurIPS 2021 Poster_

### Official Review · Reviewer_3Etu · 2021-07-08

**Rating:** 6
**Confidence:** 4

**Summary:**

This paper introduces spatiotemporal jointly decomposed convolutional ﬁlter (STDCF) to decouple spatial and temporal learning for traditional 3D convolution. The STDCF is composed of spatial atoms, joint coefficients, and temporal atoms. The STDCF parameters are optimized by the Iterative Two-Stage Learning (ITSL). Experiments show that STDCD is more efficient compared to traditional 3D CNN. The effective decomposition of 3D convolution may be of interest to the community.

**Limitations And Societal Impact:**

The authors have not stated the limitations and potential negative societal impact of their work.

The current limitation can be that the performance of STDCF has the similar performance compared to 2D CNN but more FLOPs. The inference speed test can be also added to the paper.

**Main Review:**

The idea of STDCF is decomposition for 3D convolution but the proposed operation is novel. The paper is well present and some experimental results show the effectiveness of the STDCF compared to normal rank 1 decomposition. However, I have some concerns about the potential application for STDCF and some questions on the paper as follows:

1) Algorithm 1. The proposed STDCF requires two-stage training compared to normal 3D and 2D approaches. Does this two-stage training cost more training time compared to normal methods?

2) Table 1 is somewhat difficult to highlight the useful information.
a) It seems lack the performance of using ITSL setting;
b)  Normally for Kinetics-400 dataset, 3D CNN approaches normally outperforms 2D CNN approaches. [Original R(2+1)D-ResNet34](https://arxiv.org/pdf/1711.11248v3.pdf) is able to achieve 72.0 on Kinetcs. So I suppose that R(2+1)D-ResNet50 here should get accuracy > 72.0. But here R(2+1)D is worse than TSM and TIN in which I am concerned that  R(2+1)D is not well trained in the experiment;
c) Both STDCF-R50-t and STDCF-R50-s here perform similar as TSM and TIN but contain much FLOPs even with less parameters. During inference, I suppose inference speed of the proposed method is slower than TSM and TIN;
d) Why do we need STDCF if it has the similar performance compared to 2D approaches and it might be slower than 2D CNN. Just save GPU memory?
e) As STDCF replaces 3D convolution, why not equip STDCF with SlowFast architecture to show if it is able to boost the performance or achieve similar performance but saves more GPU and requires less FLOPs.

3) Only two experiment datasets and only ResNet architecture are carried out in this study, which is not able to support the effectiveness of STDCF.



**Time Spent Reviewing:**

4

---

> ### Author Response · Authors · 2021-08-10
> **Thank you for your constructive comments!**
>
> Thank you for your careful review and constructive feedback. We address all your concerns in the following and hope the responses will alleviate your concerns.
>
> **1. Iterative learning time cost**
>
> The total training time of STDCF-R50 with ITSL with E=1 is less than full-size data training, as shown below. Note that the training times are measured on a 4 x RTX 2080ti server. As shown in the table below, one iteration of ITSL training (t+s) costs less time than full model learning.
>
> | Method      | Training time (h) |
> |:-------------:|:-------------------:|
> | STDCF-R50-t | 24.8              |
> | STDCF-R50-s | 49.7              |
> | STDCF-R50   | 98.3              |
>
>
> **2. Table 1 & Results**
>
> a)	The STDCF-R50 model learned with ITSL setting (with $E=1$) has two phases, temporal-focus learning for STDCF-R50-t, and spatial-focus learning for STDCF-R50-s. Their accuracies are shown in Table 1.
>
> b)      We will correct the results of R(2+1)D with 34-layer ResNet in the revision.
>
> c)	We test the inference speed on the validation set of Something-Somethingv1 for our full model STDCF-R50, TIN, and TSM. The test is ran on a single RTX 2080ti GPU with batchsize of 8. The resulting wall times are shown in the table below, which shows that our method has comparable inference time with TIN.
>
> |   Method  | Inference Time (s) |
> |:---------:|:------------------:|
> |   TSM [12] |        79.3        |
> |   TIN [18] |        97.6        |
> | STDCF-R50 |        95.8        |
>
>
>
> d)	We would like to clarify that our method outperforms the 2D ResNet model. Moreover, TSM and TIN can not be considered as 2D CNN, as they have modules specifically designed for processing spatiotemporal data. To show our method's superiority over 2D ResNet, please refer to the Something-Somethingv1 experiment above, we run a 2D ResNet50 model with ImageNet pretrained, and it only achieves 22.3 Top-1 accuracy. In contrast, our full STDCF-R50 model achieves 45.9 Top-1 accuracy. 2D methods can not achieve good performance on video tasks as they lack the ability to model spatiotemporal dependency.
>
> e)	Thanks for pointing out one potential future direction of experiments. We will explore plug-and-play our STDCF filter in SlowFast.
>
>
> **3. More Results**
>
> Please refer to the results of Something-Somethingv1 in the table we reply to all reviewers.

---

> > ### Comment · Reviewer_3Etu · 2021-08-18
> > **Response to authors' feedback**
> >
> > Thanks for authors' feedback. From the Re. (c), we can find the inference time for STDCF-R50 is about 20\% more compared to TSM, but the accuracy between STDCF-R50 is very close to TSM. It is clear that the researcher would choose TSM for both performance and computation. This approach needs to be further validated in terms of efficiency and efficacy.

---

> > > ### Author Response · Authors · 2021-08-19
> > > **Response to the follow-up question**
> > >
> > > Thanks for your feedback. Below is our response, we hope it will alleviate your concern.
> > >
> > > We would like to first point out that, though our method shows comparable performance with TSM in that particular experiment, our method permits directly training from video data of large spatial or temporal dimensions, by exploiting the unique atom swapping property, as stated in Section 2.1.2. For instance, TSM could not even be trained from video clips of 8x224x224 with batchsize of 16 on a RTX 2080ti GPU with 11GB memory, due to the out-of-memory (OOM) error (it requires 14.4G memory). While our STDCF model, learned with the proposed iterative two-stage learning scheme, can be easily fit into the GPU since the maximum GPU memory usage is cut by half or more here (requires 7.7G memory in maximum). In this case, TSM can only be trained by down-sampling training videos, which usually leads to inferior performance. To validate this, we provide the comparison between TSM trained with temporally and spatially downsampled data and our STDCF model learned with the ITSL strategy for the training scenario described above.
> > >
> > > |    Method    | Training Video Size | Memory | Val Top-1 Acc. |
> > > |:------------:|:------------------:|:------:|:--------------:|
> > > | TSM          |      8x224x224     |  14.4G  |      OOM error     |
> > > | TSM w/ t-ds. |      4x224x224     |  7.4G  |      38.7     |
> > > | TSM w/ s-ds. |      8x112x112     |  3.7G  |      41.5     |
> > > |  **STDCF-R50-ITSL** |      8x224x224     |  7.7G  |      **44.3**      |
> > >
> > > From the table above, it's shown that TSM models trained with temporally downsampled (TSM w/ t-ds.) and spatially downsampled (TSM w/ s-ds.) data show significant performance drop compared to our STDCF-R50 model learned with ITSL (STDCF-R50-ITSL).
> > >
> > >
> > > Furthermore, our method allows the model to easily handle tempo variances between the training and testing clips by adjusting the dilation of temporal atoms. As shown in Figure 3, our model with dilated temporal atoms at the test time can substantially mitigate the representation
> > > discrepancy between a clip of different speeds. It is not obvious how TSM directly handles tempo variance which is ubiquitous and critical for video processing.

---

> > > > ### Comment · Reviewer_3Etu · 2021-08-20
> > > > **Thanks for the catch-up**
> > > >
> > > > Thanks for the catch-up. Memory reduction is definitely one positive side of the proposed approach, which I missed before. I suggest authors to address this importance by using some real-world applications such as edge devices for video understanding. Finally, please remember to correct 1) the result for R(2+1)D performance as I mentioned in the first response, 2) include such inference and memory stuff as a limitation, which can give audiences good hints for both strength and weaknesses of the proposed approach so it is easier for people to choose depending on their own applications.
> > > >
> > > > The authors' responses solve my concern and I decide to raise my score to 6.

---

> > > > > ### Author Response · Authors · 2021-08-21
> > > > > **Thank you!**
> > > > >
> > > > > We agree that it is a promising idea to test the memory reduction property of our model on some real-world scenarios with edge devices. We will correct the performance of R(2+1)D model, and describe more about the memory limitation for readers to have a better understanding in the revision. The source code will be released to the public.
> > > > >
> > > > > Thank you for your support for our response!

---

### Official Review · Reviewer_Taqs · 2021-07-16

**Rating:** 7
**Confidence:** 4

**Summary:**

This paper presents a spatiotemporal joint filter decomposition for 3D CNN. Compared with the common 2D+1D decomposition, which is referred to as rank-1 decomposition in the paper, the proposed decomposition has an additional joint coefficients module to encode the spatial and temporal dependency. The proposed STDCF has a larger capacity in theory than the rank-1 decomposition, and is shown to be more friendly to iterative spatial-temporal learning. Toy examples and evaluations on KTH and Kinetics dataset demonstrate the efficiency of STDCF.

**Limitations And Societal Impact:**

No obvious negative societal impact is identified.

**Main Review:**

This paper addresses the 3D CNN decomposition problem, which is quite important as it is hard to fit a full-fledged 3D CNN into modern GPU when the spatial resolution, temporal sampling rate, and batch size are at a reasonable level.

The proposed decomposition method has a larger capacity, or representation power, than the commonly used 2D+1D, or rank-1, decomposition. A toy example shown in Fig. 2 clearly demonstrates the difference. A nice feature of the proposed STDCF is that it is very friendly to two-stage spatial-temporal learning, especially when proper dilation is applied. This is due to the joint coefficient layer working as a bridge between spatial atoms and temporal atoms. The results presented in Fig.5 & Fig.6 are quite convincing.

The problems or concerns this reviewer has are:
1. Why the authors set E=1 (one iteration) when evaluating on Kinectics 400? Fig.6 shows that STDCF has a huge performance boost after training for two or three iterations. There is even no sign of saturation in accuracy. This reviewer is curious about the performance on K400 when E=3.
2. How does STDCF perform on Something-Something dataset? As many of us know, Kinetics dataset is biased towards spatial information than temporal information. This characteristic is reflected in the performance gap between STDCF-R50-t and STDCF-R50-s. But what is the case for SS dataset which relies more on temporal information? The experiments are not comprehensive.

----after rebuttal----
The authors provide additional experimental results, which meet my expectation and alleviate my concerns. I therefore raised my rating by one point to Accept.


**Time Spent Reviewing:**

5

---

> ### Author Response · Authors · 2021-08-10
> **Thank you for your supportive comments!**
>
> Thank you for your thorough review and supportive comments. We hope the responses below will alleviate your concerns.
>
> **1. More iterations on ITSL**
>
> Thanks for pointing out that results of the model learned in more iterations should be shown. Below we list more results on Kinetics-400. The number at the end of the method, e.g., STDCF-R50-t-1, shows the iteration of ITSL.
>
> | Method        | Top-1 Acc. | Top-5 Acc. |
> |:---------------:|:------------:|:------------:|
> | STDCF-R50-t-1 | 67.8       | 88.3       |
> | STDCF-R50-s-1 | 70.3       | 89.1       |
> | STDCF-R50-t-2 | 70.9       | 89.6       |
> | STDCF-R50-s-2 | 72.1       | 90.1       |
> | STDCF-R50-t-3 | 72.5       | 909        |
> | STDCF-R50-s-3 | 73.3       | 91.3       |
> | STDCF-R50     | **74.1**       | **91.7**       |
>
> **2. Results on Something-Something**
>
> Please refer to the results of Something-Somethingv1 in the table we reply to all reviewers above. The performance gap between STDCF-R50-t STDCF-R50-s does become smaller than the one in Kinetics, as the model depends more on the temporal information.

---

> > ### Comment · Reviewer_Taqs · 2021-08-21
> > **Thanks for the reply**
> >
> > The additional experimental results provided in the reply meet my expectation and alleviate my concerns. Thanks for the efforts.

---

> > > ### Author Response · Authors · 2021-08-21
> > > **Thank you!**
> > >
> > > We are glad our response alleviates your concerns. We will include all the additional experiments in responses in the revision.
> > >
> > > Thank you for your support for our response!

---

### Official Review · Reviewer_mC1w · 2021-07-17

**Rating:** 6
**Confidence:** 4

**Summary:**

This work studies the problem of decomposing 3D convolutions. Previous works only consider the spatial and temporal information separately, e.g., E3D, P3D, without jointly considering the spatiotemporal information. Therefore, this work decompose the 3D convolution into a temporal atom layer, a spatial atom layer, and a joint coefficient layer. Besides, by the atom-swapping property, two training strategies are proposed to effectively train the proposed network. Experimental results show the proposed approach outperforms the baselines in terms of memory usage, while the performance remains comparable.

**Ethical Concerns:**

No.

**Limitations And Societal Impact:**

No. It is encouraged to discuss the inferior performance in terms of accuracy.

**Main Review:**

Several issues require addressing as listed below.

1) Although the idea is interesting and improves Rank-1 3D, the goal is relatively ambiguous in the paper presentation. For example, adding joint coefficient layer captures additional relationship between spatial and temporal dimension, which usually leads to a performance improvement. Nevertheless, according to the experimental results, the performance is not improved in action recognition, e.g., 75.6 top-1 accuracy of SlowFast but only 70.3 top-1 accuracy in STDCF-R50-s. The significant improvement is in memory. It is suggested to state the goal at the beginning and interpret the idea in terms of memory to better position this paper.

2) The experiments are relatively weak in terms of the accuracy. Moreover, Table 1 only shows STDCF-R50-t and STDCF-R50-s. It is suggested to list the whole model (STDCF-R50). Moreover, it is desirable to explain the memory usage difference between STDCF-R50-t and STDCF-R50-s.

3) Figures 7 and 8 visualize the attention using Grad-Cam and activation maps. Although the results demonstrate the difference, it does not show the advantage of the proposed method since spatial-temporal CNN models usually lead to a similar result. It is suggested to show the one related to the joint coefficient layer.

4) The numbering of tables and figures is messy. Line 254 mentions Table 5 but there is no Table 5. It probably refers to Figure 5. However, Line 249 mentions Figure 5. Is it a figure or a table? Moreover, Line 298 mentions Table 9, but there is no Table 9 (or Figure 9). Similarly, Line 237 mentions Table 1 for the 3D convolution but Table 1 in fact shows the results on Kinetics-400 dataset. When click the number, it refers to the abstract. The errors hinder the readability of this paper.

**Time Spent Reviewing:**

3 hours

---

> ### Author Response · Authors · 2021-08-10
> **Thank you for your constructive comments!**
>
> We are grateful for your thorough review and constructive comments. We are glad that you appreciate the idea of this paper. We hope our responses will address your concerns.
>
> **1. The goal of the paper**
>
> The goal of this paper is to present a spatiotemporal joint decomposition of 3D convolutional filters, and further show that it enables memory-efficient Iterative Two-Stage Learning. As pointed out by the reviewer, the difference between our decomposition and Rank-1 3D is the joint coefficient layer. We would like to emphasize that the joint coefficient layer is not added heuristically. It arises from the multilinear decomposition of a 3D filter, and such preserved spatiotemporal dependency enables our decomposition to gain superiority over the Rank-1 decomposition in toy image reconstruction experiment (Figure 2), in KTH experiment (Figure 5 right), and in Kinetics experiment (Table 1, R(2+1)D vs. STDCF-R50-s). The SlowFast method gains its advantage from the two-branch design, a completely different methodology that makes its number of parameters two times of our method. SlowFast can potentially benefit from further incorporating the proposed filter decomposition.
>
> **2. Accuracies and memory usage**
>
> In the table below we provide the comparisons using Kinetics-400 dataset among the full model (STDCF-R50), STDCF-R50-t, and STDCF-R50-s. It shows that our model learns with ITSL (STDCF-R50-s) achieves comparable performance with the full model, while significantly reducing the memory cost.
>
> | Method         | Memory | Top-1 Acc. | Top-5 Acc. |
> |:----------------:|:--------:|:------------:|:------------:|
> | STDCF-R50-t    | 1.9G   | 67.8       | 88.3       |
> | STDCF-R50-s    | 3.8G   | 70.3       | 89.1       |
> | STDCF-R50      | 7.9G   | 74.1       | 91.9       |
>
>
> As for the different memory usages between STDCF-R50-t and STDCF-R50-s, it is because of the different memory reduction by downsampling temporally or spatially. For instance, if an input video with a shape of 8x224x224 is downsampled temporally, which is the case of STDCF-R50-s, the input (4x224x224) and all deep features will be downsized to 50% of the original size, leading to around 50% GPU memory usage compared with the one of the full-size input video. On the other hand, if the video is downsampled spatially, as the case of STDCF-R50-t, the input (8x112x112) and all deep features will be 25% of the original size, resulting in around 25% of the original GPU memory usage.
>
>
> **3. Visualizations**
>
> ‘…since spatial-temporal CNN models usually lead to a similar result.’, that’s in fact one of the objectives of the proposed Iterative Two-Stage Learning. When the model, either the regular spatial-temporal CNN or our STDCF, is learned from full-size videos, it will mostly be able to also focus on the discriminative regions, e.g., the arms part in Figure 7, however, by requiring a large amount of memory.
>
> Enabled by the atom swapping property, our STDCF-R50 model can be learned in a two-stage manner, while it can still learn to focus on the correct discriminative regions, as shown in Figure 7, 8 with a much smaller memory footprint. As regular spatial-temporal CNNs lack the ability of atom swapping, if such two-stage learning is applied here, the temporal knowledge will be degraded in the spatial learning phase.
>
> **4. Typos**
>
> Thanks for pointing out those typos. We will correct them in the revision.

---

> > ### Comment · Reviewer_mC1w · 2021-08-19
> > **Thank you for the reply.**
> >
> > The reply mainly addresses my concern. I believe the most misleading statement is that " Different from methods that decorrelate the spatial and temporal modeling, the proposed decomposition can still capture spatiotemporal correlations in the joint coefficients." Therefore, it is easy to assume that this paper improves the performance (e.g., accuracy) by additionally modeling the spatiotemporal correlations, which has been ignored before. However, the goal is to reducing the memory usage by additionally modeling the spatiotemporal correlations. It should be mentioned in the abstract or in the title for a better focus. Moreover, the experiment setup can be further improved to make the goal more clear. I've raised the score for one point.

---

> > > ### Author Response · Authors · 2021-08-21
> > > **Thank you!**
> > >
> > > We will revise that statement to a more precise one, and emphasize more on the memory reduction property in the abstract. The experiment setup will also be improved, and the results added in the rebuttal phase will be integrated into the revision. Moreover, we will release the source code to the public.
> > >
> > > Thank you for your support for our response!

---

### Author Response · Authors · 2021-08-10
**Thank you all for your constructive reviews!**

We thank all reviewers for their insightful and constructive comments. Below are our responses to two concerns mentioned by multiple reviewers.

**1. Experiments on Something-Somethingv1**

To further validate our method, as suggested, we conduct additional experiments on Something-Somethingv1 dataset. We apply the same model STDCF-R50 we used in the Kinetics-400 experiment, and use Kinetics for pretraining. The full model is finetuned for 50 epochs with batchsize 64, learning rate 0.001 which drops by 10 at 30-th epoch. For models learned with the ITSL strategy, they are both finetuned for 20 epochs with batchsize 64, learning rate 0.002 which drops by 10 at 15-th epoch.

| Method        | Memory | Pretrain | # frames | Val Top-1 | Val Top-5 |
|:---------------|:--------:|:----------:|:----------:|:-----------:|:-----------:|
| 2D-R50        | 5.9G   | ImageNet |     8    | 22.3      | 48.2      |
| TSN [a]        | -      | ImageNet |    16    | 19.7      | 46.6      |
| TRN [b]        | -      | ImageNet |     8    | 34.4      | -         |
| I3D [1]        | -      | Kinetics |    32    | 41.6      | 72.2      |
| I3D+GCN [c]    | -      | Kinetics | 32       | 43.3      | 75.1      |
| TIN [18]       | 7.5G   | Kinetics | 8        | 45.6      | 74.2      |
| TSM [12]       | 7.1G   | ImageNet | 8        | 44.3      | 74.2      |
| STDCF-R50-t   | **1.9G** | Kinetics | 8        | 43.7      | 72.9      |
| STDCF-R50-s   | **3.8G** | -        | 8        | 44.1      | 75.7      |
| STDCF-R50     | 7.8G   | Kinetics | 8        | **45.9**  | **78.3**   |

From the above table, we can tell that the full model, STDCF-R50, achieves competitive results on Something-Somethingv1 with other methods. Moreover, the proposed ITSL can nearly recover the full model's performance at much less GPU memory cost.

**2. Limitations**

In the first draft, our model did not show extensive results on the Kinetics-400 dataset, which resulted from limited computation resource at that time. As we manage to get more computation power afterward, we provide here more results to validate our method, including the performance of the full STDCF-R50 model and more iterations of ITSL on Kinetics-400 (see the response to Reviewer Taqs), and also the above ones on Something-Somethingv1.

[a] Wang, L., Xiong, Y., Wang, Z., Qiao, Y., Lin, D., Tang, X., & Van Gool, L. (2016, October). Temporal segment networks: Towards good practices for deep action recognition. In European conference on computer vision (pp. 20-36). Springer, Cham.

[b] Zhou, B., Andonian, A., Oliva, A., & Torralba, A. (2018). Temporal relational reasoning in videos. In Proceedings of the European Conference on Computer Vision (ECCV) (pp. 803-818).

[c] Wang, X., & Gupta, A. (2018). Videos as space-time region graphs. In Proceedings of the European conference on computer vision (ECCV) (pp. 399-417).

---

### Decision · Program_Chairs · 2021-09-27

**Decision:**

Accept (Poster)

**Comment:**

All three reviewers recommend acceptance (1 rating of 7, 2 ratings of 6).

Reviewer mC1w raises the initial rating of 6 to 7, because of the convincing additional results included in the author response. However, the Reviewer still recommends revising a claim and clarifying the contribution in the abstract and title. The experimental setup should be corrected to make the objective of the work clearer.

Reviewer Taqs asked several clarifying questions, which were well addressed in the author response. The rating was thus increased to 6.

Finally, based on author's feedback, Reviewer 3Etu acknowledges the memory reduction as an important improvement and raises the rating to 6.

The ACs concur with the acceptance recommendation made by the reviewers.